# Oxidative Stress-Mediated Repression of Virulence Gene Transcription and Biofilm Formation as Antibacterial Action of *Cinnamomum burmannii* Essential Oil on *Staphylococcus aureus*

**DOI:** 10.3390/ijms25053078

**Published:** 2024-03-06

**Authors:** Lingling Shi, Wei Lin, Yanling Cai, Feng Chen, Qian Zhang, Dongcheng Liang, Yu Xiu, Shanzhi Lin, Boxiang He

**Affiliations:** 1College of Biological Sciences and Biotechnology, National Engineering Laboratory for Tree Breeding, Key Laboratory of Genetics and Breeding in Forest Trees and Ornamental Plants, Ministry of Education, Tree and Ornamental Plant Breeding and Biotechnology Laboratory of National Forestry and Grassland Administration, Beijing Forestry University, Beijing 100083, China; shilingling2019@bjfu.edu.cn (L.S.); linwei@bjfu.edu.cn (W.L.); chenfeng1996@bjfu.edu.cn (F.C.); yuxiu@bjfu.edu.cn (Y.X.); 2Guangdong Provincial Key Laboratory of Silviculture, Protection and Utilization, Guangdong Academy of Forestry, Guangzhou 510520, China; caiyl@sinogaf.cn (Y.C.); zhangq7610@sinogaf.cn (Q.Z.); gzldc@sinogaf.cn (D.L.)

**Keywords:** *Cinnamomum burmannii* essential oil, antibacterial action, oxidative stress, virulence gene, transcriptional expression, biofilm, *Staphylococcus aureus*

## Abstract

This work aimed to identify the chemical compounds of *Cinnamomum burmannii* leaf essential oil (CBLEO) and to unravel the antibacterial mechanism of CBLEO at the molecular level for developing antimicrobials. CBLEO had 37 volatile compounds with abundant borneol (28.40%) and showed good potential to control foodborne pathogens, of which *Staphylococcus aureus* had the greatest inhibition zone diameter (28.72 mm) with the lowest values of minimum inhibitory concentration (1.0 μg/mL) and bactericidal concentration (2.0 μg/mL). To unravel the antibacterial action of CBLEO on *S. aureus*, a dynamic exploration of antibacterial growth, material leakage, ROS formation, protein oxidation, cell morphology, and interaction with genome DNA was conducted on *S. aureus* exposed to CBLEO at different doses (1/2–2×MIC) and times (0–24 h), indicating that CBLEO acts as an inducer for ROS production and the oxidative stress of *S. aureus*. To highlight the antibacterial action of CBLEO on *S. aureus* at the molecular level, we performed a comparative association of ROS accumulation with some key virulence-related gene (sigB/agrA/sarA/icaA/cidA/rsbU) transcription, protease production, and biofilm formation in *S. aureus* subjected to CBLEO at different levels and times, revealing that CBLEO-induced oxidative stress caused transcript suppression of virulence regulators (RsbU and SigB) and its targeted genes, causing a protease level increase destined for the biofilm formation and growth inhibition of *S. aureus*, which may be a key bactericidal action. Our findings provide valuable information for studying the antibacterial mechanism of essential oil against pathogens.

## 1. Introduction

Food is rich in nutrients and suitable for the growth and reproduction of pathogens. Microbial food spoilage and foodborne disease remain major issues for public health worldwide [1]. Synthetic preservatives, such as tertiary butylhydroquinone (TBHQ), butylated hydroxytoluene (BHT), and several organic acids and salts, have been widely applied in the food industry to guarantee both food safety and security, but various adverse effects (immunity suppression, teratogenicity, carcinogenicity, hypersensitivity, allergic reactions, and acute toxicity) caused by prolonged use have greatly impacted public health, which has become an important topic worldwide [1,2]. Therefore, the development of novel and effective natural antimicrobial agents is of paramount importance to assure food safety and public health.

Plant essential oil has been identified to have high and extensive biological properties (such as antimicrobial, anticancer, antiviral, and antioxidant activities) [3,4,5,6,7,8]. In recent years, plant essential oils have been applied as a new natural source of antimicrobials, food preservatives, and packaging [1,9,10,11,12], and so developing and utilizing novel natural essential oils in the food industry have become one focus of future research. Several works have shown that essential oils can cause a range of damage to bacteria, including cell membrane damage and content leakage [5,7,8,13,14,15], respiratory metabolism depression [8,14,16], redox homeostasis disruption [7,17], DNA topological change, and RNA biosynthesis [8,15,18,19]. It is also noted that food poisoning and diseases caused by foodborne pathogens have become one major threat to human health and food safety due to the secretion of enterotoxins [20,21], but the growth and reproduction of pathogens are tightly dependent on biofilm formation, and thus the prevention of pathogen biofilm formation has become an effective way to ensure food safety [22]. The regulation of bacterial biofilm formation may be involved in the coordinated expression of several virulence genes [23,24,25]. In recent years, essential oils have been identified to have an anti-biofilm effect against fungi and bacteria [18,26,27,28,29,30,31] and effective inhibition of virulence-related gene expression [8,16,26,28]. Yet, few studies have studied the cellular response of bacteria to oxidative stress and ROS accumulation induced by essential oil. Therefore, the acting mechanism of essential oil involved in the virulence attenuation and biofilm formation inhibition remains enigmatic.

The genus *Cinnamomum*, a member of the family Lauraceae, is widely distributed in Southeast Asia with notable economic value owing to its rich essential oil and medicinal utilization [32,33,34]. The essential oils of *Cinnamomum* plants have been shown to have various biological effects (such as antimicrobial, anti-inflammatory, and antitumor) [32,33,34,35,36] and are widely used in the medicine, perfume, and chemical industries and especially in the food industry as natural preservatives, antimicrobial and antioxidant agents, and flavoring agents (such as brewing chocolate, chewing gum, and liquors) [37,38,39]. Of note, *Cinnamomum burmannii* is one of the most promising sources in the food, pharmaceutical, and cosmetic industries due to its unique compounds (such as borneol, α-terpineol, and α-pinene) [34,38]. Based on our studies on different *C. burmannii* germplasms, some accessions were selected with a high yield of essential oil (1.2–1.6%) and an average proportion (38.7%) of borneol, and notably, we established a standard system for the utilization of *C. burmannii* [40,41]. Yet, the antibacterial mechanism of *C. burmannii* leaf essential oil (CBLEO) is unknown, which has intercepted CBLEO application in modern industry.

The aim of this work was to unravel antibacterial action and to highlight the molecular mechanism that governs bacterial growth and biofilm inhibition for developing CBLEO as a potential source of natural antibacterial agent. To this end, one plus tree of *C. burmannii* (accession CB01) was used to detect the volatile profile of CBLEO and to assess antibacterial activity on seven representative foodborne pathogens, and the antibacterial action of CBLEO on *Staphylococcus aureus* (susceptible strain) was observed. As an initial step toward exploring the antibacterial action of CBLEO, we focused on the dynamic effects of CBLEO on bacterial growth, content release, electric conductivity, ROS and MDA formation, protein oxidation, and cell morphology in *S. aureus* subjected to different doses (1/2×MIC, 1×MIC, and 2×MIC) and times (0–8 h). Such an assay could help to unravel how CBLEO induces oxidative stress. To highlight the antibacterial acting mechanism of CBLEO on *S. aureus* at the molecular level, a comparative assay was conducted on the association of ROS accumulation with some key virulence-related gene transcription, protease level, biofilm formation, and the interaction with genome DNA in the response of *S. aureus* to CBLEO at different levels and times. This study presents the effect of essential-oil-mediated oxidative stress and ROS accumulation on the transcription of virulence-associated regulators as an attempt to elucidate the antibacterial mechanism of essential oil.

## 2. Results

### 2.1. Identification of Volatile Compounds in C. burmannii Leaf Essential Oil (CBLEO)

To develop *C. burmannii* leaf essential oil (CBLEO) as a potential source for application, we detected the chemical compounds of CBLEO by GC-MS. A total of 37 compositions were identified (Table 1), accounting for 99.56% of total oil. The main chemical components of CBLEO mainly included 14 monoterpene hydrocarbons, 29 sesquiterpene hydrocarbons, 6 monoterpene alcohols, 6 sesquiterpene alcohols, and 5 sesquiterpene hydrocarbons, of which monoterpene (85.38%) was the dominant group of components. The richest compound was borneol (28.40%), followed by bornyl acetate (9.43%), eucalyptol (9.22%), D-limonene (7.44%), α-pinene (3.96%), β-cymene (3.96%), β-caryophyllene (3.71%), α-terpineol (3.15%), α-phellandren (2.67%), sabinene (2.53%), β-myrcene (2.41%), β-pinene (2.38%), and camphor (2.14%). These results revealed a complex of chemical components with rich borneol in CBLEO.

### 2.2. Assay of Antibacterial Activity of CBLEO

The values of DIZ, MIC, and MBC were detected to determine the antibacterial activity of CBLEO. In the case of CBLEO, the DIZ values ranged from 7.51 to 28.72 mm across all tested strains, and the MIC and MBC values were in the range of 1.0−16.0 μg/mL and 2.0−32.0 μg/mL, respectively (Table 2), emphasizing that CBLEO had good inhibition effects on foodborne pathogens. Of note, the maximum value of DIZ (28.72 nm) and the minimum values of MIC (1.0 μg/mL) and MBC (2.0 μg/mL) were all marked in the response of *S. aureus* to CBLEO, indicating a strong bactericidal effect of CBLEO on *S. aureus*. Hence, our following work was focused on the exploration of the antibacterial action of CBLEO on *S. aureus* in order to develop CBLEO as a potential antibacterial agent.

### 2.3. Effect of CBLEO on Bacterial Growth of S. aureus

To unravel the antibacterial action of CBLEO against *S. aureus*, the assay of the antibacterial kinetics curve was conducted on the CBLEO treatments with five different concentrations (1/8×MIC, 1/4×MIC, 1/2×MIC, 1×MIC, and 2×MIC) and times (0–24 h), all of which showed a dose/time-dependent inhibition manner for bacterial growth (Figure 1). The greatest inhibitory effect was recorded at 2×MIC for 24 h with the number of viable cells reduced by 96.86% from 7.0 to 0.2 lg CFU/mL, followed by a 93.57% decrease at 1×MIC, but a 38.29%, 20.28%, and 6.86% decline was marked for 1/2×MIC, 1/4×MIC, and 1/8×MIC, respectively. A notable inhibition of bacterial growth was identified within the 1 h incubation of 1×MIC and 2×MIC, and a complete inhibition occurred within the first 8 h and 12 h at 2×MIC and 1×MIC, respectively. Yet, the control cells showed a normal growth status (Figure 1). These results revealed a great bactericidal potential of CBLEO toward *S. aureus*.

### 2.4. Impact of CBLEO on Cell Structure of S. aureus

The damage to cell structure (cell wall and membrane) in CBLEO-treated *S. aureus* was evaluated as part of an attempt to understand the antibacterial action of CBLEO. Firstly, the effect of CBLEO on cell wall damage was tested by the leakage assay of AKP (cell-wall-damage-biomarker enzyme). In the presence of CBLEO, AKP activity in *S. aureus* suspensions increased by a dose/time-dependent pattern (Figure 2A). After 8 h of exposure to CBLEO, AKP activity at 2×MIC was 1.7- and 0.8-fold greater than that at 1/2×MIC and 1×MIC, respectively, and notably increased activity was recorded within the first 2 h at both 1×MIC and 2×MIC, but the control exhibited no change in AKP activity (Figure 2A), so it was concluded that CBLEO could cause the destruction of the cell wall of *S. aureus*.

Next, CBLEO-mediated damage to the permeability of the cell membrane was assayed by detecting electric conductivity. During exposure to CBLEO, a dose/time-dependently increased profile for electric conductivity was marked in *S. aureus* suspensions, of which a rapid increase was detected within the first 1 h (Figure 2B). After the 8 h incubation of CBLEO, the values of electric conductivity at 1/2×MIC, 1×MIC, and 2×MIC were 4.0-, 7.3-, and 8.9-fold greater than those of the control (Figure 2B), indicating that CBLEO could effectively cause the cell membrane damage and permeability increase of *S. aureus* with the intracellular electrolyte release.

As for cell membrane integrity, the releases of intracellular protein and nucleic acid were tested. After the incubation of *S. aureus* with different doses of CBLEO (1/2×MIC, 1×MIC, and 2×MIC), the released amount of protein showed a dose/time-dependent increase, in which the increased degree of protein release at 2×MIC for 8 h was 2.0- and 0.7-fold greater than at 1/2×MIC and 1×MIC, respectively (Figure 2C). A similar dose/time-dependent increase pattern was also noted for nucleic acid leakage (Figure 2D). Yet, those in the control showed no significant change (Figure 2C,D). These results indicated that CBLEO could result in irreversible damage to the cytoplasmic membrane integrity of *S. aureus* with a loss of cellular materials.

To highlight the antibacterial action of CBLEO on *S. aureus*, we further performed SEM analysis to explore the influence of CBLEO on the cell morphology of *S. aureus*. The untreated *S. aureus* cells retained a normal and complete appearance with a smooth surface, intact cell membrane, and cell wall structure (Figure 3A), whereas the CBLEO-treated *S. aureus* cells became irregular, and cell surface collapsed or shriveled (Figure 3B). We thus considered that CBLEO could change cell morphology and damage the cell membrane of *S. aureus*. This further confirmed the above assayed results of CBLEO-caused damage to the permeability and integrity of the cell membrane of *S. aureus* (Figure 2).

### 2.5. Effect of CBLEO on Cellular MDA and ROS Generation and Protein Oxidation of S. aureus

Considering that CBLEO could damage cell structure and control the bacterial growth of *S. aureus* (Figure 1, Figure 2 and Figure 3), it was vital to determine the potential of CBLEO to induce oxidative stress destined for the cell membrane damage and growth inhibition of *S. aureus*. Firstly, MDA was selected as a lipid peroxidation biomarker to evaluate the temporal amount change in *S. aureus* cells during exposure to CBLEO. A dose/time-dependent increase pattern was identified for intracellular MDA (Figure 4A), of which MDA content was much higher at 2×MIC than at both 1×MIC and 1/2×MIC after 8 h exposure, while the control showed almost no change, implying that CBBLEO could induce the cell lipid peroxidation of *S. aureus*. This allowed us to explore the intracellular ROS generation and protein oxidation of *S. aureus* cells as an oxidative stress indicator. The amounts of intracellular ROS and protein carbonyl formation increased in *S. aureus* cells in a dose/time-dependent manner during exposure to CBLEO, both of which showed the maximum value after 8 h of exposure to CBLEO at 2×MIC, but no notable change was detected for the control (Figure 4B,C), emphasizing that CBLEO-induced oxidative stress could result in an accumulation of ROS and protein oxidation product in *S. aureus* cells.

### 2.6. Effect of CBLEO on Cellular Total Protein Concentration of S. aureus

Given the effect of CBLEO-induced oxidative stress on the cellular lipid peroxidation and protein oxidation of *S. aureus* (Figure 4B,C), it was essential for us to investigate whether CBLEO-induced oxidative stress affected the total protein of *S. aureus* cells. Hence, a dynamic analysis of cellular total protein was performed in *S. aureus* exposed to CBLEO. Compared with the control, the amount of cellular total protein exhibited a dose/time-dependently decreased profile in *S. aureus* exposed to CBLEO, and a 25.79%, 34.37%, and 53.01% decline was observed at 1/2×MIC, 1×MIC, and 2×MIC after the 8 h incubation of CBLEO, respectively (Figure 4D), suggesting the induction of bacterial protein fragmentation or its biosynthesis disturbance by CBLEO.

### 2.7. Effect of CBLEO on Cellular Biofilm Development and Protease Activity of S. aureus

Another concern was whether CBLEO shows anti-biofilm activity against *S. aureus*. A dose/time-dependent decline pattern was noted for biofilm biomass in *S. aureus* cells exposed to CBLEO, of which the reduction in biofilm biomass was 43.64%, 63.75%, and 85.42% after 8 h of exposure to 1/2×MIC, 1×MIC, and 2×MIC, respectively, but the increased biofilm formation was observed for the control (Figure 5A), emphasizing that CBLEO had a good anti-biofilm effect on *S. aureus*. It was also noted that the biofilm formation decline was highly associated with protease production [42]. Compared with biofilm biomass (Figure 5A), the protease activity exhibited a dose/time-dependent increase in *S. aureus* subjected to CBLEO, while a decrease in protease activity was detected for the control (Figure 5B), indicating that the CBLEO-induced low capacity of biofilm formation may be associated with an increase in protease activity.

### 2.8. Effect of CBLEO on Genome DNA of S. aureus

It was reported that carvacrol or citral from essential oils could be chimeric with bacterial DNA and break the DNA structure of *S. aureus* and *E. coli* [8,14,16]. The same single electrophoretic band of genomic DNA was marked for *S. aureus* from all CBLEO-treated and control samples (Appendix A), suggesting no direct effect of CBLEO on the genome DNA of *S. aureus*.

### 2.9. Effect of CBLEO on Transcript of Virulence-Related Genes and Regulatory Proteins in S. aureus

The above findings that oxidative stress could effectively induce ROS accumulation (Figure 4B) and biofilm formation reduction (Figure 5A) with no direct effect on bacterial genome DNA (Appendix A) in the response of *S. aureus* cells to CBLEO prompted us to highlight the mechanism for how CBLEO causes virulence attenuation and biofilm formation inhibition at the molecular level. Biofilm formation, one key virulence determinant, is controlled by several virulence genes in response to ROS. To ascertain the anti-biofilm mechanism of CBLEO at the molecular level, some vital virulence-associated genes responsible for biofilm formation, including *agrA* (accessory gene regulator A), *sigB* (sigma factor B, involved in biofilm formation and stress response), *sarA* (staphylococcal accessory regulator A), *cidA* (murein hydrolase regulator, involved in cell lysis and extracellular DNA release), *icaA* (intercellular adhesin A, involved in cell wall and biofilm formation), and *rsbU* (involved in the regulation of sigB and biofilm formation), were selected as potential antibacterial targets to analyze dynamic transcription changes in *S. aureus* cells during exposure to CBLEO by qRT-PCR detection.

As shown in Figure 6, the transcriptional levels of *sigB/rsbU/agrA/sarA/icaA/cidA* in CBLEO-treated *S. aureus* cells were all down-regulated in a dose/time-dependent pattern, of which the lowest transcript level was recorded for 8 h at 2×MIC, but an increase in them was marked in the control, as also noted for biofilm formation in CBLEO-treated *S. aureus* cells (Figure 5A), indicating that CBLEO could inhibit the transcription of virulence-related genes destined for the reduction in *S. aureus* biofilm formation. Of note, the down-regulated degree of the transcriptional level of *sigB* (88.02%) and *rsbU* (92.01%) was much higher than that of *agrA/sarA/icaA/cidA* (68.89−73.97%) after 8 h of exposure at 2×MIC (Figure 6), implying that SigB and RsbU may be a crucial regulator to control biofilm formation in CBLEO-treated *S. aureus* cells.

Together, the CBLEO-induced collaborative transcription repression of virulence-related genes was mainly responsible for the inhibition of biofilm formation, of which both RsbU and SigB may be the antibacterial targets of CBLEO against *S. aureus* (Figure 7).

## 3. Discussion

### 3.1. Rich Volatile Profiling with High Borneol Amount and Good Antibacterial Activity of CBLEO

Plant-derived essential oils have been widely used as natural antimicrobials in the food industry [9,11,12]. In this work, 37 volatile compositions were identified in *C. burmannii* leaf essential oil (CBLEO) (Table 1), of which many compounds and their contents were different from those reported for other *Cinnamomum* species (such as *C. pauciflorum*, *C. zeylanicum*, and *C. camphora*) [26,37,43], indicating a difference in volatile profiling and its contents among different *Cinnamomum* species. It was also noted that most of our detected volatile compounds have been shown to have high antimicrobial activity, especially some major constituents, such as borneol (28.31%) [44,45,46], D-limonene (7.44%) [5,6,47,48], α-pinene (3.96%) [49], β-Caryophyllene (3.71%) [4,48,50], and α-terpineol (3.15%) [36]. Combined with the good inhibitory effect of CBLEO on all tested foodborne pathogens (Table 2), it seems certain that these chemical compositions may be a potential of CBLEO to control pathogens. Several studies have indicated that borneol shows several pharmacological activities including analgesic, anti-inflammatory, and antioxidant properties [45,46]. Our finding of borneol (28.31%) as the richest compound of CBLEO (Table 1) that was higher than that for *C. burmannii*, *C. zeylanicum*, *C. camphora*, *C. pauciflorum*, and *C. tamala* (0.81–11.95%) [34,37,43,51,52] revealed that it may be the most promising antibacterial agent. Altogether, the advantage of CBLEO over other essential oils from different *Cinnamomum* species was its rich volatile components with a high proportion of borneol, emphasizing that CBLEO may be a novel source for utilization. Of note, *S. aureus* was marked as the most susceptible pathogen (Table 2), and thus the following work focused on highlighting the antibacterial action of CBLEO against *S. aureus* for the development of CBLEO as a natural antibacterial agent for potential utilization.

### 3.2. ROS-Generation-Mediated Oxidative Stress and Cell Membrane Damage Involved in Antibacterial Action of CBLEO

During the exposure of bacteria to the essential oil, lipophilic components could bind to the bacterial cell surface and penetrate the outer membrane, and then its lipid bilayer made contact with the hydrophobic part of the cell membrane, subsequently causing a toxic effect, leading to cell wall and membrane damage, function destruction, material release, ROS generation, and cell death [5,7,8,14,16]. Yet, the essential-oil-mediated mechanism of bacterial cell membrane damage is still unclear. Accumulating evidence showed that intracellular material leakage may be a good biomarker of irreversible damage to the cell membrane [5,7,13]. Here, a close negative correlation was established between the inhibitory effect of bacterial growth (Figure 1) and a significant increase in material release, extracellular AKP activity (cell-wall-damage-marker enzyme), and electric conductivity in *S. aureus* during exposure to CBLEO (Figure 2), implying that the destruction of cell membrane structure induced by CBLEO may be one pivotal cause for the growth inhibition of *S. aureus*. This was in line with the previously found antibacterial effect of essential oil on several foodborne pathogens (such as *Bacillus subtilis*, *B. cereus*, *Escherichia coli*, *E. faecalis*, *Listeria monocytogenes*, *Pseudomonas aeruginosa*, *Shigella dysenteries*, *S. aureus*, *Salmonella typhimurium*, and *Shigella flexneri*) [5,7,8,14,15].

ROS-induced lipid peroxidation during oxidative stress is known as one key initiator that causes cell membrane damage. MDA, the most abundant product of lipid peroxidation, can induce cellular protein oxidation [53] and thus widely serves as an indicator of oxidative stress for studying microbial growth, cell death, and disease incidence [7,54]. However, the association of oxidative stress with MDA variation in bacterial cells caused by essential oil remains unclear. Given that a similar accumulative pattern of MDA (Figure 4A) and ROS (Figure 4B) in CBLEO-treated *S. aureus* cells was positively associated with the degree of bacterial growth inhibition, material release, and cell membrane damage (Figure 1, Figure 2 and Figure 3), it seems certain that CBLEO-induced oxidative stress may be one crucial bactericidal effect. Our results were consistent with the effect of the essential oils from different plants and natural compounds (such as carvacrol, citral, flavonoid, anthocyanin, dihydromyricetin, and eugenol) on *S. flexneri*, *S. aureus*, *E. coli*, and *P. digitatum* [7,8,14,16,17,19,28,55,56] and also evidenced by previous studies showing that the effects of antibacterial agents (such as fluconazole, cerulein, catechin, chitosan, miconazole, indomethacin, and hypocrellin A) could cause oxidative stress with the consequence of ROS accumulation and membranous damage destined for bacterial cell death [44,57,58,59,60].

In general, many microbes possess a range of defensive systems to detoxify ROS [61]. The thioredoxin (Trx) and glutaredoxin (Grx) systems are the two major thiol-dependent antioxidant systems in the defense against the oxidative stress of bacteria cells [62]. The Trx system, which is composed of NADPH, thioredoxin reductase (TrxR), and Trx, can provide the electron to thiol-dependent peroxidases (known as peroxiredoxins, Prx) to remove reactive oxygen species such as glutathione peroxidase (GPx) and contribute to the redox state of methionine sulfoxide reductases (Msr) for the repair of oxidized proteins [62,63]. The Grx system, containing NADPH, glutathione reductase (GR), GSH, and Grx, is involved in the defense against oxidative stress via the efficient removal of various ROS by GPx [64]. It is also noted that catalase (CAT), superoxide dismutase (SOD), and TrxR homolog alkyl hydroperoxide peroxidase subunit C/F (Ahpc/f) participate in the antioxidant process in bacteria [62]. Several works have shown that the mutation of *grx1/2/5*, *trx1/2*, *msr1/2*, *trxR*, *grl*, *ahpC*, or *gpx3* in some bacteria (such as *E. coli*, *Helicobacter pylori*, *Mycobacter tuberculosis*, *Streptococcus pyogenes*, *S. cerevisiae*, or *S. aureus*) was sensitive to oxidative stress with a decreased survival [62,65,66,67,68,69,70]. In combination with our findings of cell membrane damage, bacterial growth inhibition, and MDA and ROS accumulation in CBLEO-treated *S. aureus* (Figure 1, Figure 2 and Figure 4A,B), it was considered that the CBLEO-induced deficiency of enzymatic antioxidant systems may be one critical antibacterial factor. This could be verified by our recent results that essential-oil-mediated decrease in ROS-detoxified enzymes (SOD, CAT, Prx, GPx, and GST) may likely contribute to the cell membrane damage and growth inhibition of *S. flexneri* [7] and that exogenous ROS scavenger (such as SOD and CAT) treatment could alleviate ROS formation and the membrane damage of *P. digitatum* and *S. aureus* cells [17,60]. Therefore, CBLEO may act as an inhibitor of the enzyme systems participating in antioxidant responses in *S. aureus*.

Also noteworthy was the impact of ROS accumulation on cellular protein oxidation [71]. Here, the increased amount of cellular protein carbonyl in CBLEO-treated *S. aureus* (Figure 4C) was positively associated with the accumulated amount of ROS and MDA (Figure 4A,B), both of which showed a negative correlation with the declined content of total cellular protein (Figure 4D). Thus, we concluded that CBLEO-induced oxidative stress could cause cellular protein oxidation and biosynthesis disturbance, as also noted for the effect of essential oil on *P. digitatum* and *S. aureus* [14,17].

Together, CBLEO-mediated oxidative stress and ROS accumulation may be the primary driver of the cell membrane damage of *S. aureus*. Yet, little attention has been paid to the cross-talk between oxidative stress and virulence-associated factor expression required for biofilm formation during the exposure of bacteria to the essential oil.

### 3.3. CBLEO-Induced Transcription Repression of Virulence-Associated Genes in S. aureus as Pivotal Antibacterial Action

Foodborne disease has become a serious issue affecting human health and food safety. *S. aureus*, one of the most common foodborne pathogens, can grow in various foods and cause food poisoning by secreting enterotoxins that cause various disease symptoms (such as nausea, vomiting, and diarrhea) [20,21] and thus poses a serious threat to human health [22]. *S. aureus* enterotoxins, one superfamily of secreted virulence factors, are generally regulated by a quorum-sensing *agr* system via the autoinducer peptide (AIP) and two divergent transcripts (called RNAII and RNAIII), of which the RNAII transcript is an operon of *agr* genes (*agrBDCA*) that encode key factors for *agr* regulatory activation [72]. Of these, AIP is produced from the AgrD precursor and then processed and exported as a quorum signal by AgrB to activate sensor kinase AgrC and response regulator AgrA, subsequently leading to the induction of the *agr* system and the upregulation of RNAII/RNAIII transcription essential for virulence production [72,73,74]. Also, AgrA is known as a key virulence regulator, but SarA has been identified as a positive regulator of *agr* activity in *S. aureus* [23,24,25]. Yet, it is unclear whether essential oil can affect bacterial AgrA or SarA activity. Here, the coordinately repressed transcriptions of both *sarA* and *agrA* in the response of *S. aureus* cells to CBLEO (Figure 6A,B) were temporally and positively correlated with bacterial growth inhibition (Figure 1), indicating that the CBLEO-induced repression of the transcriptions of *sarA* and its targeted *agrA* may contribute to the growth inhibition of *S. aureus*, which was the case for the effect of essential oil on *C. albicans*, *C. violaceum*, *P. aeruginosa*, *P. arotovorum*, *P. aroidearumor*, and *S. aureus* [16,18,26,27,28]. AgrA and SarA, two key global regulators of virulence genes, were tightly controlled by transcription regulator SigB [24,75], which may in turn regulate the transcriptions of several virulence-related factors crucial for the cell processes (such as stress response and biofilm formation) of *B. subtilis*, *L. monocytogenes*, *P. aeruginosa*, and *S. aureus* [25,75,76]. Considering that a similar dose/time-dependently repressed transcription of *sigB/agrA/sarA/icaA/cidA* in CBLEO-treated *S. aureus* cells (Figure 6) was temporally and positively correlated to the inhibition of bacterial growth (Figure 1) and biofilm formation (Figure 5A) but concomitantly with an increase in protease activity (Figure 5B), it seems likely that the CBLEO-mediated coordinate repression of the *sigB/agrA/sarA/icaA/cidA* transcript could activate the expression of the protease gene destined to increase its production in *S. aureus* cells during exposure to CBLEO, which may contribute to the inhibition of biofilm formation and bacterial growth. This was consistent with the antibacterial effect of essential oil on *E. coli*, *A. baumannii*, *C. violaceum*, *C. albicans*, *P. aroidearumor*, *P. aeruginosa*, and *P. arotovorum* [8,18,26,27,28,29,31]. In support of our results, *sigB*, *sarA*, *agrA*, *icaA*, or *cidA* mutation in *S. aureus* could increase protease amount with a significant decline of biofilm formation [42,77,78,79,80,81,82,83], and the exogenous addition of protease notably limited the biofilm formation of *S. aureus* [84,85].

Also of note was the role of RsbU in SigB activity activation [86,87]. The mutation of *rsbU* could repress the transcription of *sigB* and its targeted downstream gene (*ica/sarA/agr*) related to bacterial biofilm formation [76,88]. In this work, the reduced biofilm formation (Figure 5A) was positively associated with the coordinately repressed transcription of *rsbU/sigB/agrA/sarA/icaA/cidA* during the exposure of *S. aureus* cells to CBLEO (Figure 6), and notably, the transcriptions of *rsbU* and *sigB* were significantly down-regulated (Figure 6C,F) and thus revealed that both RsbU and SigB may be key regulators in controlling the biofilm formation of *S. aureus* cells exposed to CBLEO. This fact was supported by previous results that *rsbU* or *sigB* mutation could increase the expression of the protease gene and decrease biofilm formation [85,87]. Therefore, it seems that CBLEO-mediated transcriptional repressions of RsbU and SigB may be pivotal antibacterial targets against *S. aureus*. Yet, the actual mechanism by which essential oil repressed the transcription of virulence regulators destined for biofilm formation inhibition remains unknown.

ROS-mediated oxidative stress could cause AgrA oxidation to loss regulatory activity [23,25], and the mutation of *rsbU*, *sigB*, *agrA*, or *sarA* in *S. aureus* increased susceptibility to oxidative stress and inhibited biofilm formation [23,85]. In this work, a close correlation was established between the massive accumulation of ROS and protein oxidation (Figure 4B,C), the low transcript of rsbU/sigB/agrA/sarA/icaA/cidA (Figure 6), the less formation of biofilm (Figure 5A), and the notable inhibition of bacterial growth (Figure 1) in *S. aureus* exposed to CBLEO, indicating that the CBLEO-induced growth inhibition of *S. aureus* may be attributed mostly to the limitation of biofilm formation via the depression of the transcription of the virulence-related regulator caused by ROS accumulation, which may confer the key acting mechanism of CBLEO on *S. aureus*. This finding was compatible with the antibacterial effect of essential oil on *P. aeruginosa*, *C. violaceum*, and *E. coli* [8,18] and was also consistent with previous results that antibacterial agents (linezolid, benzimidazole, and vancomycin) could cause oxidative stress, leading to the transcription repression of the virulence-related gene destined for protease activity increase and biofilm formation inhibition [85,89,90]. All of this indicated that ROS-mediated oxidative stress could be a critical initiator of the transcriptional repression of key regulators (RsbU and SigB) and their targeted genes in the response of *S. aureus* to CBLEO, leading to a biofilm formation decrease, which eventually causes bacterial growth.

Altogether, the collaborative transcription repression of virulence-related genes and the effective increase in protease activity, as well as the notable inhibition of biofilm formation, may respond specifically to an increase in ROS-induced oxidative stress in *S. aureus* by CBLEO. Of these, the RsbU/SigB-mediated transcription regulatory system was responsible for the bactericidal effect of CBLEO against *S. aureus*.

## 4. Materials and Methods

### 4.1. Plant Materials

The leaves of *C. burmannii* were collected from one 10-year-old plus tree with high borneol content (germplasm accession CB01) planted in our research base (Guangdong Huaqingyuan Biotechnology Co., Ltd.) in Meizhou City, Guangdong Province (E115°50′1″, N24°28′28″), China [40].

### 4.2. Extraction of Essential Oil and Analysis of Chemical Components of CBLEO

Essential oil was extracted by steam distillation, and the fresh leaves (about 50 g) were powdered and subjected to hygro-distillation using a modified Clevenger-type apparatus for 5 h [91]. The obtained oils were collected, measured, and dried with anhydrous Na_2_SO_4_ and then stored in a sealed tube at −20 °C for further use. The extracted essential oils were weighted, and their yield (1.61%) was calculated and expressed as the percentage (%, *w*/*w*) of fresh leaf [7].

The chemical constituents of CBLEO were detected by GC-MS method. The obtained oil sample was performed on GCMS-QP2020W/O gas chromatograph (Shimadzu, Kyoto, Japan), equipped with SH-R×ITM-5SIL MS column (30 m × 0.25 mm, 0.25 μm) [91]. The temperature program was as follows: from 70 °C to 160 °C at 2 °C/min and hold for 2 min, then increased to 220 °C at 10 °C/min and kept for 5 min. The carrier gas was nitrogen at 1.19 mL/min. The GC inlet was set in a splitting mode with split ratio of 1:20 and at 230 °C, and 1.0 μL of diluted samples (1/10, *v*/*v*, in hexane) was injected. The quadrupole MS operating parameters: interface temperature 200 °C; electron impact ionization at 70 eV with scan mass range of 45−450 *m*/*z*. The volatile compounds were identified according to the Mass Spectral Library database and retention indices of authentic reference standards.

### 4.3. Bacterial Strains and Culture

Seven representative foodborne pathogens used for antibacterial activity assay were purchased from the China Center of Industrial Culture Collection (CICC), including four Gram-negative bacteria of *Pseudomonas aeruginosa* (CICC 21636), *Escherichia coli* (CICC 10389), *Salmonella enterica* subsp. *enterica* (CICC 10982), and *Enterobacter aerogenes* (CICC 10293) and three Gram-positive bacteria of *Staphylococcus aureus* (CICC 10384), *Bacillus subtilis* (CICC 10275), and *Listeria monocytogenes* (CICC 21633).

To assess the antibacterial activity of CBLEO on foodborne pathogens, the tested bacterial suspensions were prepared in nutrient broth (NB) (Difco 234000, Becton, Dickinson and Company, Franklin Lake, NJ, USA), except for *L. monocytogenes* which was prepared in brain heart infusion (BHI) broth (Difco 237500, USA), and incubated at 37 °C for 24 h. Each strain inoculum was suspended in 0.85% of sterile saline to obtain a standard microbial density (about 10^7^ CFU/mL).

### 4.4. Assessment of Antibacterial Activity of CBLEO

#### 4.4.1. Detection of Diameter of Inhibition Zone (DIZ)

The detection of DIZ by agar disc diffusion was used to screen antimicrobial activity of CBLEO [7]. The prepared sterile media of BHI agar for *L. monocytogenes* and NB agar for other bacteria were cooled to 50 °C and solidified in the sterilized plate, and each strain inoculum (100 µL, 10^7^ CFU/mL) was streaked over on the surface of culture medium using commercial bacteriological loops, and then the sterile 6 mm paper disc impregnated with CBLEO (10 μL, 50 mg/mL) was placed on medium surface. After the incubation at 37 °C for 24 h, antimicrobial activity was determined by a clear zone around the disc, and the DIZ was detected. Ampicillin (10 μg/disc, 10 mg/mL) and sterile distilled water (10 μL) were used as the positive and negative controls, respectively.

#### 4.4.2. Determination of Minimum Inhibitory (MIC) and Bactericidal Concentration (MBC)

The values of MIC and MBC of CBLEO were detected by microdilution method [7]. A series of two-fold dilutions of CBLEO (0.125−32 μg/mL) was added into the bacteria suspension (1 × 10^7^ CFU/mL) in the wells of a sterile microplate and cultured overnight at 37 °C. The lowest concentration of CBLEO that showed no visible bacteria growth was defined as the MIC, and the MBC was expressed as the lowest concentration of CBLEO required to kill bacteria. Ampicillin was applied as reference antibacterial agent.

### 4.5. Analysis of Bacterial Growth Kinetics of S. aureus

Antibacterial kinetics assay was used to assess antibacterial mechanism of CBLEO on *S. aureus* (the most susceptible strain) [5]. In order to increase CBLEO solubility, the stock CBLEO was prepared in 4% dimethyl sulphoxide (DMSO) [50] to obtain five different doses (1/8×MIC, 1/4×MIC, 1/2×MIC, 1×MIC, and 2×MIC) and then was added into 100 mL of bacterial suspension (10^7^ CFU/mL). After incubating for 3, 6, 9, 12, 15, 18, 21, and 24 h, the collected suspension was used to detect optical density (OD) at 600 nm by ultraviolet spectrophotometer (Agilent Cary 3500, Agilent, Santa Clara, CA, USA). The growth kinetic curve of *S. aureus* was elaborated by drawing the lg number of CFU/mL versus incubated time [7].

### 4.6. Analysis of Antibacterial Mechanism of CBLEO on S. aureus

#### 4.6.1. Cell Membrane Permeability

The impact of CBLEO on cell membrane permeability of *S. aureus* was evaluated by detecting electric conductivity using electrical conductivity meter (DDS-11D, Shanghai, China) [7]. The obtained *S. aureus* cells by centrifugation (10,000 rpm) at 4 °C for 10 min were washed with 5% glucose until electric conductivity was close to that of 5% glucose and defined as isotonic bacteria for electric conductivity detection, and then different doses of CBLEO were respectively added. After incubation at 37 °C for 0, 1, 2, 4, 6, 8, 10, 12, and 24 h, the electric conductivity of the mixtures was detected and marked as EC_2_. The bacteria in 5% glucose treated in boiling water for 5 min and 5% glucose containing different doses of CBLEO were used as positive and negative controls, respectively, and their electric conductivities were measured and marked as EC_0_ and EC_1_, respectively. Cell membrane permeability was expressed as the relative electric conductivity and calculated by the following equation: Relative electric conductivity (%) = [(EC_2_ − EC_1_/EC_0_) × 100].

#### 4.6.2. Integrity of Cell Membrane

The membrane integrity of *S. aureus* cells was evaluated by detecting the leakage of intracellular nucleic acid and protein [7]. *S. aureus* cells (1 × 10^7^ CFU/mL) were incubated at 37 °C with CBLEO at different levels (1/2×MIC, 1×MIC, and 2×MIC) and times (0–8 h), and the samples were collected at different times. After centrifugation (10,000 rpm) at 4 °C for 10 min, the supernatants were used to detect the amounts of nucleic acid and protein by using ultraviolet spectrophotometer (Agilent Cary 3500, Agilent, Santa Clara, CA, USA). The detected result of nucleic acid was expressed as the absorbance value at 260 nm (OD_260_), and the amount of protein was standardized to the used amount of *S. aureus* cells (μg/mg).

#### 4.6.3. Cell Wall Damage

The impact of CBBLEO on cell wall damage of *S. aureus* was analyzed by detecting release of alkaline phosphatase (AKP) [92]. Different doses (1/2×MIC, 1×MIC, and 2×MIC) of CBLEO were added into *S. aureus* cells (10^7^ CFU/mL) and incubated at 37 °C, and then the supernatants collected at different times (0–8 h) were used for the detection of AKP activity by commercial kit (RS0904F, Redshineen Biotech, Guangdong, China). The AKP activity was standardized to the used amounts of *S. aureus* cells, and the result was expressed as U/L.

#### 4.6.4. Scanning Electron Microscope (SEM) Analysis

The impact of CBLEO on cell morphology of *S. aureus* was tested by SEM assay [16]. The *S. aureus* suspension (10^7^ CFU/mL) was added into CBLEO (1×MIC), and the group with equal amount of absolute ethanol was applied as the control. After 2 h of incubation, the collected cells were washed with phosphate-buffered saline buffer and then fixed in glutaraldehyde (2.5%) at 4 °C for 12 h, followed by dehydration under the different levels of ethanol gradient (30, 50, 80, 90, and 100%). The specimens were dried at critical point of CO_2_ and coated with gold-palladium by Polaron E5100 II (Polaron Instruments Inc., Hatfield, CA). Finally, the samples were observed with a scanning electron microscopy (SEM, JSM-7001F, JEOL, Tokyo, Japan) at voltage of 10 kV.

#### 4.6.5. Analyses of Cellular Protein Oxidation and ROS and MDA Production

Oxidative stress and lipid peroxidation of *S. aureus* induced by CBLEO were assessed by detecting the amounts of ROS generation and protein oxidation (two key markers of oxidative stress) and MDA accumulation (one biomarker of lipid peroxidation) [7,71]. The *S. aureus* cells (10^7^ CFU/mL) were incubated with different doses (1/2×MIC, 1×MIC, and 2×MIC) of CBLEO at 37 °C, and the samples were collected respectively at different times (0−8 h). After centrifuging at 10,000 rpm for 10 min under 4 °C, the obtained cell pellets were used to determine protein carbonyl, ROS, and MDA by using assay kits of ab126287, ab113851, and ab118970 (Abcam, Shanghai, China), respectively. The amount of ROS was given in fold of untreated controls, and the content of protein carbonyl (protein oxidation product) was standardized to the used amount of *S. aureus* cells and defined as nmol/mg. The MDA content was standardized to the used amounts of *S. aureus* cells, and the result was expressed as nmol/mg.

### 4.7. Analysis of Bacterial Total Protein

The *S. aureus* cells (10^7^ CFU/mL) were incubated with CBLEO at 37 °C under different doses of 1/2×MIC, 1×MIC, and 2×MIC and times (0−8 h), and the samples were respectively collected to centrifugate at 10,000 rpm for 10 min under 4 °C. The obtained cell pellets were used for detection of total protein amount by using BCA Protein Assay Kit (102536, Abcam, Shanghai, China). The result was standardized to the used amounts of *S. aureus* cells and expressed as μg/mg.

### 4.8. Total Protease Assay

Total protease activity was detected in the above-obtained cell pellets of *S. aureus* by analysis kit (ab111750) according to the manufacturer’s instructions, and the fluorescein isothiocyanate (FITC)-labeled casein was used as a general substrate. The protease activity was standardized to the used amounts of cell total protein and expressed as U/mg protein using BSA as standard. One unit (U) was defined as the amount of protease that cleaves substrate to yield an amount of fluorescence equivalent to 1.0 μmol of unquenched FITC per min at 25 °C.

### 4.9. Analysis of Virulence-Associated Gene Expression 

Total RNA of *S. aureus* was extracted by RNAprep Cell/Bacterial kit (Tiangen, Beijing, China) and was reverse-transcribed by PrimeScriptTM RT reagent kit (Takara, Osaka, Japan). qRT-PCR was performed on BIO-RAD CF× ConnectTM Real-Time System using SYBR Green qPCR Mix (Biomarker, Beijing, China). All amplified primers used for the detection of virulence genes are listed in Appendix A, and 16S rRNA was used as internal reference. The relative expression value of target genes in comparison with reference gene was counted by 2^−ΔΔCt^ method [8], and the expression level in *S. aureus* from the treatments of different doses of CBLEO at 0 h was arbitrarily set to 1.00 for standardization.

### 4.10. Assay of Anti-biofilm Activity

The impact of CBLEO on biofilm formation of *S. aureus* was assessed by a microtiter plate assay [18]. An overnight culture of *S. aureus* was added into a 96-well dish containing 200 µL of LB broth supplemented with various levels of CBLEO. After incubation at 37 °C for 1−8 h, the bacterial cells were washed with PBS to remove all unattached cells and media components, and then 250 µL crystal violet staining solution (0.1%) was added and incubated for 20 min at 25 °C. After this, the plates were washed with PBS buffer 2−3 times and solubilized with ethanol. The absorbance at 570 nm (OD_570_) was detected in CBLEO-treated (different concentrations) or control sample at 0−8 h, of which the value of OD_570_ at 0 h was arbitrarily set to 1.00 for standardization, and the result was defined as fold change.

### 4.11. Assay of Binding Activity of CBLEO to Bacteria Genome DNA 

The binding activity of CBLEO to genomic DNA of *S. aureus* was tested by agarose gel electrophoresis method [93]. Genomic DNA of *S. aureus* was isolated by TIANamp Bacteria DNA Kit (Tiangen, Beijing, China), and the value of OD_260_ was measured to calculate DNA concentration. An aliquot (100 ng) of genome DNA was incubated with CBLEO at different doses (1/2×MIC, 1×MIC, 2×MIC, 4×MIC, 8×MIC, and 16×MIC) at 37 °C for 0.5, 1, 3, and 5 h under darkness, and each incubated mixture (5.0 μL) was loaded to run agarose gel electrophoresis. The bacteria treated by PBS was used as the control.

### 4.12. Statistical Analysis

All the results were recorded as mean ± SD (standard deviation) three independent replicates and analyzed by ANOVA employing Student’s test at *p* < 0.05. All statistical assays were performed by IBM SPSS Statistics 25 software.

## 5. Conclusions

In this work, *C. burmannii* leaf essential oil (CBLEO) was identified to have diverse volatile compounds and a high amount of borneol, as well as good antibacterial activity, and *S. aureus* was the most susceptible pathogen. CBLEO could act as a strong inducer for ROS accumulation and the oxidative stress of *S. aureus*, causing cell structure damage and giving rise to the effective repression of virulence-related gene transcription with significant inhibition of biofilm formation destined for the growth inhibition of *S. aureus*. Notably, the comparative association among ROS accumulation, virulence-associated gene transcription, protease production, biofilm formation, and bacterial growth in *S. aureus* across different doses and times of CBLEO treatment led to the identification of RsbU and SigB as important transcriptional regulators crucial for bacterial biofilm formation and growth inhibition. The RsbU/SigB-mediated repression of biofilm formation caused by CBLEO-oxidative stress may be a key antibacterial target against *S. aureus*. Our findings should provide valuable information for those studying the acting mechanism of essential oil against pathogens. Further research should focus on the exploration of the functional attributes (especially for borneol) of essential oil from *C. burmannii* leaf in food practical utilization as a natural antibacterial agent.

## Figures and Tables

**Figure 1 ijms-25-03078-f001:**
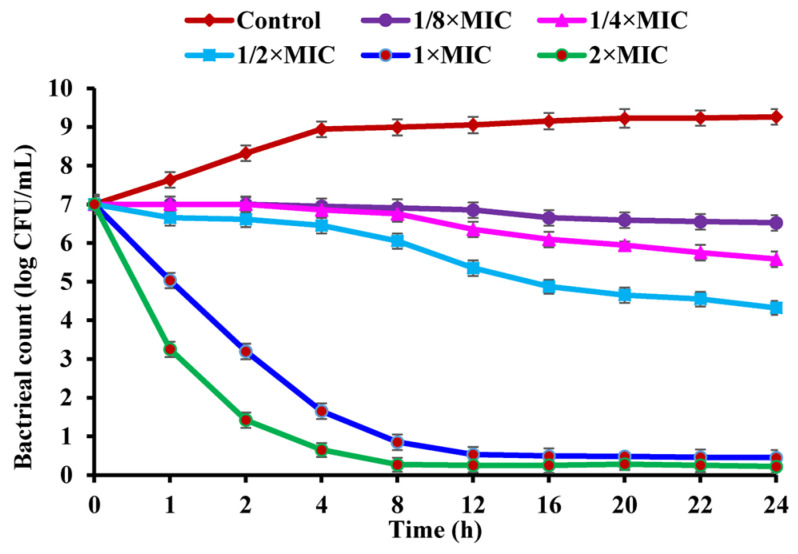
The growth kinetics curve of *Staphylococcus aureus* affected by CBLEO. The data represent the mean value ± SD of three parallel replicates.

**Figure 2 ijms-25-03078-f002:**
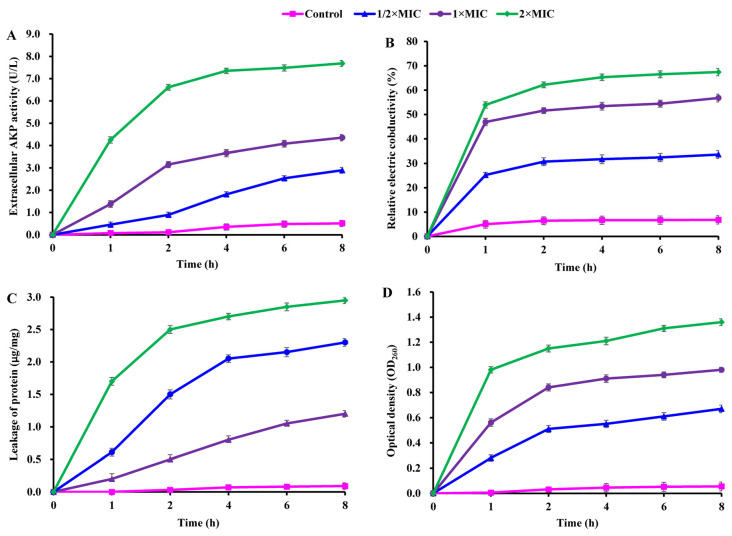
Effect of CBLEO on cell wall and cell membrane of *S. aureus*. (**A**) Extracellular activity of alkaline phosphatase (AKP); (**B**) relative electric conductivity; (**C**) leakage of protein; (**D**) release of 260 nm absorbing material. Data represent the value ± SD of three replicates.

**Figure 3 ijms-25-03078-f003:**
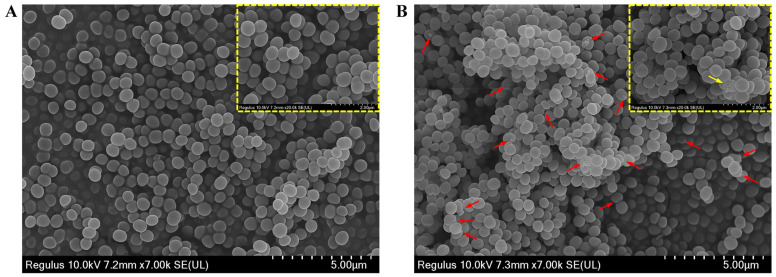
Effect of CBLEO on cell morphologically of *S. aureus* by scanning electron microscope (SEM) assay. (**A**) SEM image of untreated *S. aureus*; (**B**) SEM image of *S. aureus* treated with CBLEO (1×MIC) for 2 h. Red arrows represent cell morphology change and cell membrane damage. Yellow boxes represent the details of cell morhology, bar = 2 μm. Yellow arrow represents the severely damaged cell.

**Figure 4 ijms-25-03078-f004:**
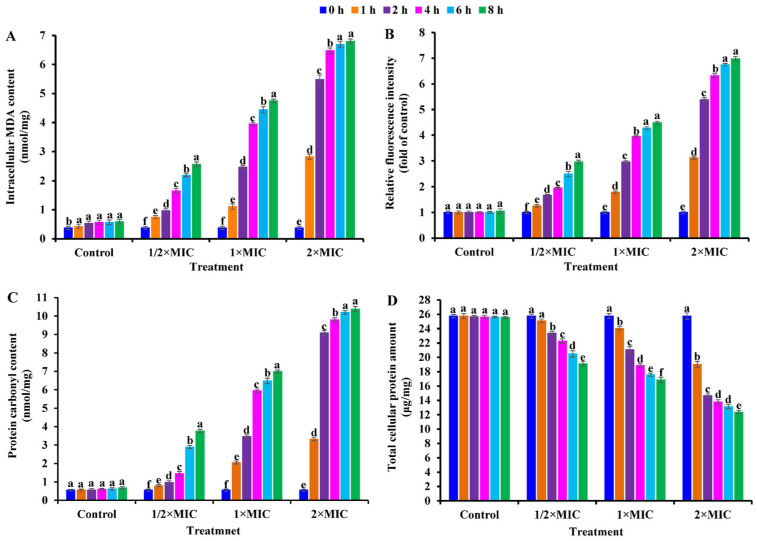
Effect of CBLEO on cell lipid peroxidation and oxidative stress in response of *S. aureus* to different doses and times. (**A**) Intracellular mallondialdehyde (MDA) level; (**B**) intracellular ROS generation; (**C**) protein carbonyl content; (**D**) total cellular protein. Data represent mean value ± SD of three parallel replicates, and different letters denote significant differences (*p* < 0.05).

**Figure 5 ijms-25-03078-f005:**
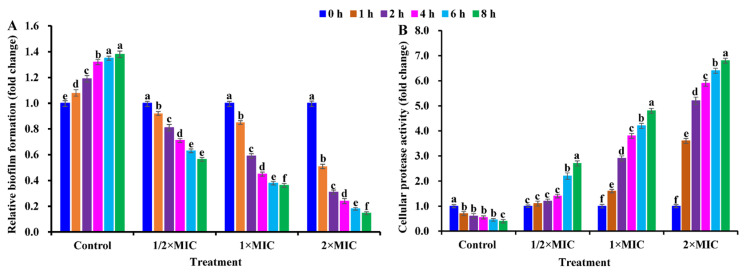
Effect of CBLEO on biofilm formation and protease activity in response of *S. aureus* cells to different doses and times. (**A**) Assessment of inhibitory capacity of CBLEO on biofilm formation by microtiter plate assay; (**B**) change in cellular protease activity. Values of biofilm formation and protease production in *S. aureus* cells from control and CBLEO-treated samples with different doses at 0 h were arbitrarily set to 1.00 for standardization. Data represent mean value ± SD of six parallel replicates, and different letters denote significant differences (*p* < 0.05).

**Figure 6 ijms-25-03078-f006:**
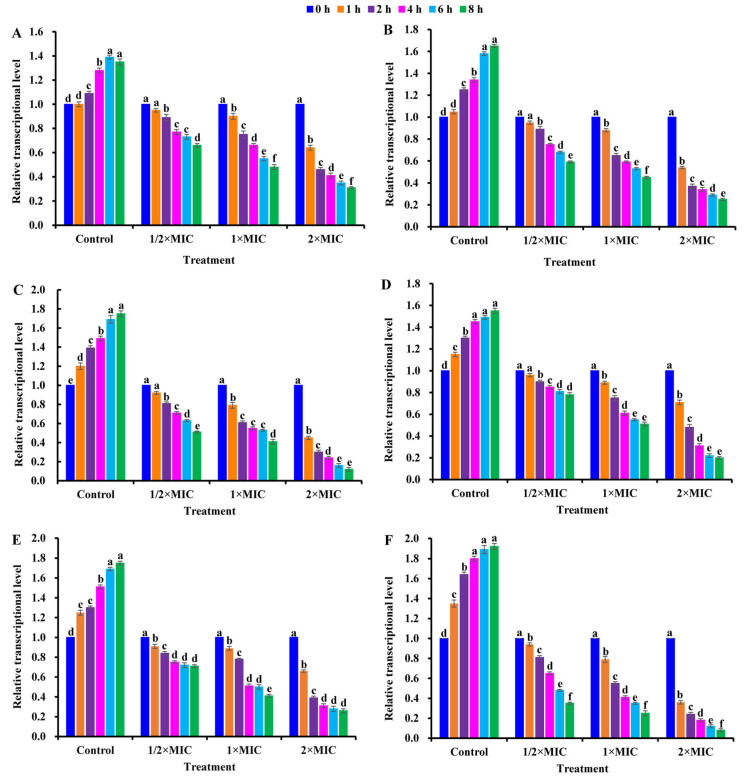
Effect of CBLEO on transcriptions of virulence-associated regulators of *S. aureus* under exposure to different concentrations and times by qRT-PCR detection. (**A**) Relative transcription of *agrA* (accessory gene regulator A); (**B**) relative transcription of *sarA* gene (staphylococcal accessory regulator A); (**C**) relative transcription of *sigB* (sigma factor B); (**D**) relative transcription of *icaA* (intercellular adhesin A); (**E**) relative transcription of cidA gene (encoding for holin); (**F**) relative transcription of *rsbU* (SigB activator). Relative expression values were counted as 2^−ΔΔCt^, and 16S RNA was used as internal control. Transcription level in *S. aureus* cells from control and CBLEO-treated samples with different concentrations at 0 h was arbitrarily set to 1.00 for standardization. Data represent mean value ± SD of three parallel replicates, and different letters denote significant differences (*p* < 0.05).

**Figure 7 ijms-25-03078-f007:**
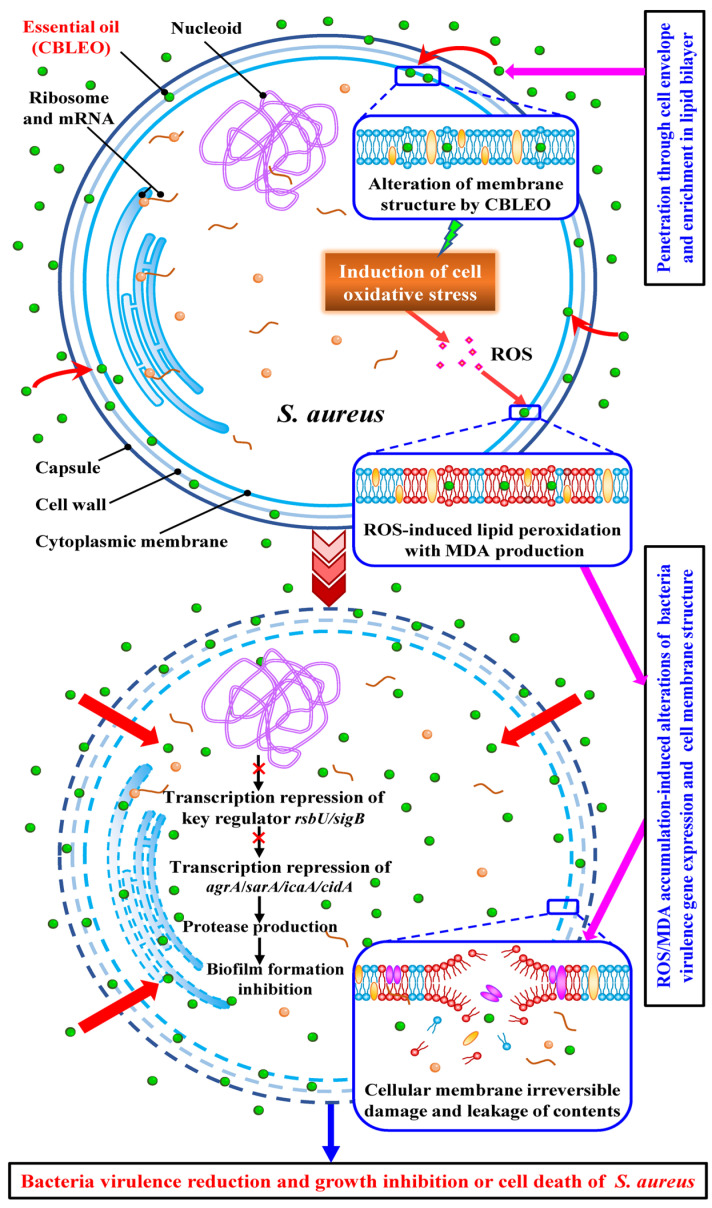
The antibacterial acting mode of *C. burmami* leaf essential oil (CBLEO) on *S. aureus*. The identified antibacterial mechanism of CBLEO against *S. aureus* is based on the present work and is summarized from the two perspectives of virulence-related gene transcription regulation and cellular structure destruction. The abbreviations are shown as follows: *agrA*, accessory gene regulator A; *sarA*, staphylococcal accessory regulator A; *sigB*, sigma factor B; *icaA*, intercellular adhesin A; ROS, reactive oxygen species; MDA, mallondialdehyde.

**Table 1 ijms-25-03078-t001:** Compounds and contents of essential oils from *Cinnamomum burmami* leaves by GC-MS analysis.

Type	Compound	Formula	RI ^A^	Percentage (%) ^B^
Monoterpene hydrocarbons (14)	α-Thujene	C_10_H_16_	902	0.56 ± 0.03 ^e^
	α-Pinene	C_10_H_16_	948	3.96 ± 0.17 ^bc^
	(−)-Camphene	C_10_H_16_	943	2.03 ± 0.12 ^c^
	Sabinene	C_10_H_16_	897	2.53 ± 0.11 ^c^
	β-Pinene	C_10_H_16_	943	2.38 ± 0.10 ^c^
	β-Myrcene	C_10_H_16_	958	2.41 ± 0.10 ^c^
	α-Phellandrene	C_10_H_16_	969	2.67 ± 0.12 ^c^
	3-Carene	C_10_H_16_	948	0.32 ± 0.02 ^e^
	4-Carene	C_10_H_16_	919	0.19 ± 0.01 ^f^
	β-Cymene	C_10_H_14_	1042	3.96 ± 0.19 ^bc^
	D-Limonene	C_10_H_16_	1018	7.44 ± 0.25 ^b^
	β-*cis*-Ocimene	C_10_H_16_	976	0.43 ± 0.02 ^e^
	γ-Terpinene	C_10_H_16_	998	0.50 ± 0.03 ^e^
	α-Terpinolen	C_10_H_16_	1052	0.96 ± 0.06 ^e^
Monoterpene alcohol (6)	Eucalyptol	C_10_H_18_O	1059	9.22 ± 0.29 ^b^
	Linalool	C_10_H_18_O	1082	0.63 ± 0.03 ^e^
	Borneol	C_10_H_18_O	1138	28.40 ± 0.62 ^a^
	Terpinen-4-ol	C_10_H_18_O	1137	1.50 ± 0.09 ^cd^
	α-Terpineol	C_10_H_18_O	1143	3.15 ± 0.18 ^bc^
	Guaniol	C_10_H_18_O	1228	0.23 ± 0.01 ^f^
Monoterpene ketone (1)	Camphor	C_10_H_16_O	1121	2.14 ± 0.10 ^bc^
Monoterpene aldehyde (1)	α-Citral	C_10_H_16_O	1174	0.19 ± 0.01 ^f^
Monoterpene ester (2)	Bornyl acetate	C_12_H_20_O_2_	1277	9.33 ± 0.24 ^b^
	Geranyl acetate	C_12_H_20_O_2_	1352	0.25 ± 0.01 ^f^
Sesquiterpene hydrocarbons (5)	β-Caryophyllene	C_15_H_24_	1494	3.71 ± 0.19 ^bc^
	α-Caryophyllene	C_15_H_24_	1579	0.99 ± 0.05 ^e^
	Germacrene D	C_15_H_24_	1515	0.58 ± 0.03 ^e^
	α-Guaiene	C_15_H_24_	1469	0.19 ± 0.01 ^f^
	Germacrene B	C_15_H_24_	1603	1.74 ± 0.10 ^cd^
Sesquiterpene alcohol (6)	Elemol	C_15_H_26_O	1522	0.42 ± 0.02 ^e^
	*trans*-Nerolidol	C_15_H_26_O	1564	1.36 ± 0.07 ^cd^
	Spathulenol	C_15_H_24_O	1536	1.72 ± 0.08 ^cd^
	Guaiol	C_15_H_26_O	1614	1.28 ± 0.08 ^cd^
	(−)-Spathulenol	C_15_H_24_O	1536	0.51 ± 0.03 ^e^
	Bulnesol	C_15_H_26_O	1614	0.44 ± 0.02 ^e^
Sesquiterpene ester (1)	Caryophyllene oxide	C_15_H_24_O	1507	1.00 ± 0.08 ^cd^
Others (1)	Cinnamyl acetate	C_11_H_12_O_2_	1367	0.24 ± 0.01 ^f^
Total				99.56 ± 0.21

^A^ Retention index of present experiment determined on SH-R×ITM-5SIL MS column using n-alkanes (C_8_–C_25_) series. ^B^ Relative percentage was calculated by peak area, and data represent average of three replicates (*n* = 3). Different lowercase letters represent significant differences (*p* < 0.05).

**Table 2 ijms-25-03078-t002:** Assay of antibacterial activity of *C. burmannii* leaf essential oil (CBLEO) against seven representative foodborne pathogens.

Microorganisms	CBLEO			Antibiotic ^A^		
	DIZ ^B^ (mm)	MIC ^B^ (μg/mL)	MBC ^B^ (μg/mL)	DIZ ^B^ (mm)	MIC ^B^ (μg/mL)	MBC ^B^ (μg/mL)
Gram-positive bacteria						
*Bacillus subtilis*	21.31 ± 0.65 ^b^	2.0	4.0	40.70 ± 0.71 ^a^	0.125	0.25
*Listeria monocytogenes*	13.33 ± 0.45 ^bc^	4.0	8.0	35.31 ± 0.56 ^b^	0.125	0.25
*Staphylococcus aureus*	28.72 ± 0.72 ^a^	1.0	2.0	30.05 ± 0.56 ^b^	0.5	1.0
Gram-negative bacteria						
*Escherichia coli*	9.71 ± 0.61 ^bc^	8.0	16.0	37.52 ± 0.61 ^a^	0.25	0.5
*Pseudomonas aeruginosa*	7.51 ± 0.48 ^bc^	16.0	32.0	37.31 ± 0.57 ^a^	0.25	0.5
*Enterobacter aerogenes*	10.21 ± 0.51 ^bc^	8.0	16.0	33.71 ± 0.61 ^b^	0.25	0.5
*Salmonella*	9.02 ± 0.43 ^bc^	8.0	16.0	39.51 ± 0.45 ^a^	0.125	0.25

^A^ Ampicillin was used as reference antibacterial agent. ^B^ DIZ: diameter of inhibition zone; MIC: minimal inhibitory concentration; MBC: minimal antibacterial concentration. The values are mean of triplicate detection (*n* = 3) ± standard deviation. Different lowercase letters represent significant differences (*p* < 0.05).

## Data Availability

All data presented in this study are included in the published article and Appendix A. Further inquiries can be directed to the corresponding author.

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
