# Peer review of "Oxidative Stress-Mediated Repression of Virulence Gene Transcription and Biofilm Formation as Antibacterial Action of *Cinnamomum burmannii* Essential Oil on *Staphylococcus aureus"

_ijms, 2024, doi:10.3390/ijms25053078_

Round 1

Reviewer 1 Report

Comments and Suggestions for Authors

Check the attached file

Comments on the Quality of English Language

English should be improved

Reviewer 2 Report

Comments and Suggestions for Authors

Undoubtedly, the article is of interest to a wide range of readers and opens up both fundamental and practical areas for further research.

But some mistakes were found:

Abstact –  21 «.. of which Staphylococcus aureus had the greatest inhibition  zone diameter (28.72 nm)»   - zone diameter must be in mm.

Salmonella (CICC 10982) -    add spp name- Salmonella enterica subsp. enterica CICC 10982

472 –«the tested bacterial  suspensions were prepared in nutrient broth (NB), expect for brain heart infusion (BHI)» -  add full name of the nutrient media and their manufacturers.

479 «Each strain inoculum (100 µL, 107 CFU/mL) was coated on the culture medium  surface (BHI for L. monocytogenes, NB for other bacteria)». For the disc -diffusion method agar-based media are used - they are not specified.

Reviewer 3 Report

Comments and Suggestions for Authors

Round 2

Reviewer 1 Report

Comments and Suggestions for Authors

Its fine

Comments on the Quality of English Language

Minor improvement needed

Reviewer 3 Report

Comments and Suggestions for Authors

The authors have successfully addressed most of the issues raised. However, they failed to expand on the enzyme systems that might have been affected by treatment with CBLEO. No more experiments are demanded, only discussion. The enzyme systems participating in antioxidant responses in bacteria consist of the thioredoxin and glutaredoxin systems, superoxide dismutases, methionine sulfoxide reductases and peroxiredoxins (related to the thioredoxin system). We suggest the authors mention that these systems (or other that the authors may consider appropriate), might have been affected by CBLEO which could have acted as an inhibitor.

Round 3

Reviewer 3 Report

Comments and Suggestions for Authors

The authors have adresssed all points raised.